# Copper technology in the Arabah during the Iron Age and the role of the indigenous population in the industry

**David Luria***

Tel Aviv University, Tel Aviv, Israel

* dudiluria1@gmail.com

## Abstract

Following the Egyptian withdrawal in the mid-12th century BCE from their involvement in the Arabah copper production, and after an additional period of organization, the degree of copper efficiency and production at Timna and Faynan increased in the Early Iron Age (11th–9th centuries), rendering the region the largest and most advanced smelting centre in the Levant. The existing paradigm offered as an explanation for this technical and commercial success is based on extraneous influence, namely, the campaign of Pharaoh Sheshonq I near the end of the 10th century BCE that spurred a renewed Egyptian involvement in the Arabah copper industry. An alternative paradigm is suggested here, viewing the advances in Arabah copper technology and production as a linear development and the outcome of continuous and gradual indigenous improvements on the part of local craftsmen, with no external intervention. Behind these outstanding technical achievements stood excellent managerial personnel, supported by an innovative technical team. They employed two techniques for copper-production optimization that can be defined based on concepts taken from the world of modern industrial engineering: (i) "trial and error", in which the effect of each production variable was tested individually and separately, and (ii) "scaling-up", in which the size of some production elements (i.e., tuyère) was increased by using existing techniques which required minimum developmental costs and experimental risks.

## Introduction and background

Large-scale copper production in the Arabah is known from extensive excavations and surveys that focused mainly on the sites of Timna on the western side of the valley, in Israel, and Faynan, ca. 100 km to the north on the eastern side of the valley, in Jordan [1,2]. Recently, the area of Wadi Amram, south of Timna, has been explored as well [3–5]. The rich data accumulated pertains both to the archaeological and to technological aspects of the copper production.

In the Late Bronze Age (LBA), the operations at Timna were part of a wider industrial establishment controlled by Egypt from the end of the 14th until the mid-12th centuries BCE [6–8]. Following the Egyptian withdrawal from Canaan in the 12th century BCE, local copper production in the Arabah not only continued into the Early Iron Age (11th–9th centuries BCE), but also expanded to include the site of Faynan, reaching an unprecedented scale,

**Data availability statement:** All relevant data are within the manuscript and its Supporting Information files. Full information for the

specific specimens considered for this study (including specimen numbers, repository information, and geographic locations) are provided within their original publication, Ben-Yosef et al. 2019, including Supporting Information (S1 Table). No new samples were examined for the sake of this study. No permits were required for the described study, which complied with all relevant regulations.

**Funding:** No specific funding was received for this work.

**Competing interests:** The authors have declared that no competing interests exist.

particularly at the latter site: the quantity of slag produced from 1200–1150 BCE was only ca. 1,600 tons; from 1100–1050 BCE this gradually increased to ca. 5,600 tons, and from 1000–950 BCE to ca. 15,600 tons. The peak of production was achieved during 900–850 BCE, at ca. 23.000 tons. This widespread operation ceased quite abruptly towards the end of the 9th century BCE [9,10].

The existence paradigm of the organization of production of the Arabah copper industry assumes that Egypt played a central technological and organizational role in the Late Bronze Age and then again, after a period of absence in ca. 925 BCE. This paradigm assumed that there had been no major change or innovation following the departure of the Egyptians in the 12th century BCE, with local craftsmen utilizing existing 'Egyptian' LBA technology, which underwent minor improvements until a major technological leap occurred in the late 10th century due to the renewed regional involvement of the Egyptians of the 22nd Dynasty, in wake of the Sheshonq I (Shishak) campaign. This led to the introduction of innovations in technology and production organization that resulted in a significant increase in the efficiency of copper production [11], The existing paradigm was purportedly supported by the discovery of a scarab of Sheshonq I at Faynan [12], Furthermore, the advocates of the existing paradigm suggested that Egypt became involved not only in the technical part of the copper industry, but also exercised direct influence on the organization of production and trade at this time [11].

Ben-Yosef et al. 2019 [11] attempted to explain the assumed technological 'leap' caused by the extraneous intervention of the Egyptians following Sheshonq I's campaign through the lens of the theory of punctuated equilibrium. This theory [13] assumes stasis periods followed by brief intervals of rapid evolutions. Thus, they suggest that the technological leap in the Arabah occurred: "After generations of internal efforts to better the technology—with limited success—the techno-social system was receptive of extraneous influences that facilitated the same cause" [11].

However, a close look at the involvement of the Egyptians shows that while there is no question of their present at Timna during the latter part of the LBA, their exact role in Faynan remains unclear. It has been shown [14,15] that the Hathor shrine (Site 200) was added on top of and incorporated into an existing Semitic shrine, suggesting that the Egyptians took control of the local, ongoing, copper-production activity, but were not its initiators. Avner et al. [5] also suggested that the indigenous people were the geologists, the mining engineers and the physicists behind this industry.

This article questions the assumed critical impact of these external factors on the Arabah copper production in the Early Iron Age. An alternative scenario is proposed, emphasizing indigenous processes in which local craftsmen utilized high levels of engineering and managerial skills acquired and improved over years of continuous production, resulting in the huge success of this industry. The main sites discussed in this paper are shown in Fig 1.

## The archaeometallurgical data

**"Production Systems (PS)".** Ben Yosef et al. [11, Fig 5] assess copper-production quality in the Arabah from the LBA to the end of the 9th century BCE and define four stages in the development of smelting, designated "Production Systems" (PS) (see Fig 2).

Ben Yosef et al. assume that the first stage, PS 0 (1300–1140 BCE), relates to the LBA and refers to the small-scale production at Timna under Egyptian control, briefly noted above. At this time, activities in the Faynan area were limited; evidence of moderate copper production found only in the deepest layers of the excavations at KEN Area M (Layer M5) were considered "ephemeral and opportunistic".

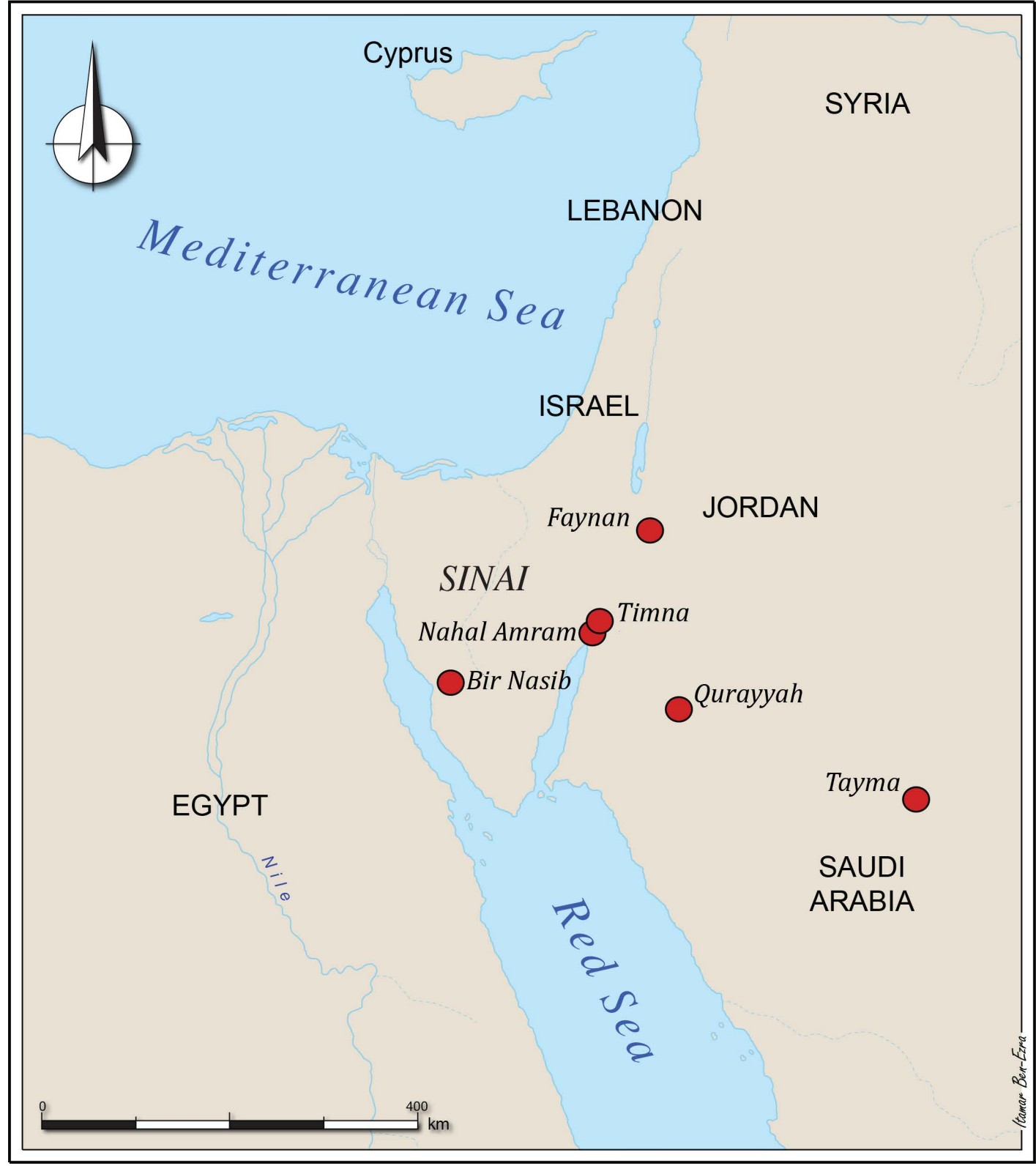

**Fig 1. Map with sites mentioned in the text.**

**Fig 2. Average Cu content in slag samples plotted against their radiocarbon dates.** Taken as is from Ben-Yosef et al. [11, Fig 3], the three distinct groups (marked by the rectangular frames) demonstrate the gradual development from PS I, PS II, and PS III. The dashed lines at 1140 and 925 BCE represent historical events: The Egyptian withdrawal in 1140 BCE and the Sheshonq I campaign in 925 BCE. Ken = Khirbet en-Nahas, KAJ = Khirbet al-Jariye, T = Timna.

The second stage, PS I (1140–1000 BCE), following the Egyptian withdrawal, relates to a prolonged period that witnessed an increase in the efficiency in rate of copper production, as well as the beginning of a process of centralization, in which the cooperation between Timna and Faynan was established.

The third stage, PS II (1000–925 BCE), relates to a continuation of the gradual technological improvement. Thus, both PS I and PS II show continuous and gradual improvement, but they differ in regards to archaeological considerations, namely, the investment in fortifications that were established at Khirbet en-Nahas in this phase.

The fourth stage, PS III (925–830 BCE), relates to a major reorganisation of the industry and the most advanced stage of the technical development, in which new, larger furnaces and tuyères were utilized and a change of flux was initiated at Timna, from Fe-oxides to the more effective Mn-oxides [11]. In addition, production was centralized at fewer sites—Timna, where only one smelting site (Site 30) was active, while three sites operated at Faynan (Khirbat en-Nahas, Faynan 5 and Barqa al-Hetiye). This high level of standardization and centralization led to increased internal production dependence and is one possible explanation for the ultimate simultaneous collapse of the Arabah copper production system towards the end of the 9th century [16].

**The evidence for technical efficiency and standardization: Residual copper content.** Evidence for the improvement of technical standardization during these four stages is shown in Fig 2, which presents the average residual copper content within slag attained from 143 samples, along with the Standard Deviation (STD); decrease of STD means a high degree of standardization that reflects better skill in the monitoring of the production process, and *vice versa*.

The results of PS O include only one site at Timna, T3, dated to LBA, showing a relatively high-quality production of copper, with 1.1% of residual copper in the slag (n = 45) and a high level of standardization (STD = 0.77%) [17, p. 52–53]. The results of PS I show a residual copper content of 1.65% Cu ± 0.87% (n = 56). Throughout the PS I–PS II stages, the production rate increased significantly (see [18, p. 93] for calculation of slag amounts), followed by a gradual improvement in efficiency, evidenced by the residual copper in PS II (n = 46) being reduced to 1.03% and the STD decreased to 0.57%. The mean copper content measured in the PS III slag dropped to 0.51%, but more importantly, the STD decreased to a new low of 0.18%.

The improvement in efficiency and standardization of copper production over the course of the Early Iron Age (11th–9th centuries BCE) was accompanied by additional technical advancements, including the initiation of slag crushing for the extraction of copper prills, i. e., copper-iron droplets trapped within the slag material. [10, p. 734, 831], and the secondary smelting of Early Bronze Age slag that served for exploitation of an additional available copper source [19]. It was also suggested that the changes in efficiency may be the result of changing the flux from iron oxides to manganese.

**An alternative approach to the technological stages in copper production.** The distinction between the Production Systems described above involved mainly archaeological considerations, some unrelated to the technology, such as building activity at Khirbet e-Nahas, as well as historical events [11]. However, in this unique case the focused should be given to technology development, mostly carried out by a limited number of personnel. Therefore, in our view, the developmental stages should be based only on technological considerations. Such a consideration yields a division into two main Technological Phases: TP I and TP II. Table 1 shows the correlation between the two proposed "Technological Phases" to the paradigm of the four "Production Systems". TP I (= PS O—PS II) basically includes the technology used at Timna during the LBA, which continued to exist until the change in the technological tool kit in the 10th century BCE. TP II (= PS III) is the final and most advanced production phase.

The residual copper concentrations in the slag samples from the Arabah sites and their average dates (as presented above in Fig 2, as shown in ref. [11]) were subjected to Cluster Analysis. The statistical procedure selected here was K-means [20], and the number of clusters were determined using the Elbow Method. The outcome is shown in Fig 3. It presents three clusters: Clusters 1 and 2 include samples that are related to the first Technological Phase (TP I), whereas Cluster 3 correlates to the second Technological Phase (TP II).

The two clusters included in TP I present a complex development from the end of LBA and throughout the early Iron Age, showing that following the withdrawal of the Egyptians from Timna, there was a large decrease in efficiency, from ca. 1.1% Cu to 2.5% Cu, as indicated in Cluster 1 and the lowest point of Cluster 2. This phenomenon can be explained by the expansion of the industry to include Faynan–a step that resulted in decreased control over the mining and smelting activities, and possibly a shortage in trained personnel at this initial stage. Thus, the duration of Cluster 1 can be defined as an adaptation phase. Thereafter, the duration of Cluster 2 (ca. 1050–960 BCE) was characterized by continuous and gradual improvement in efficiency.

Ben-Yosef et al. [11] postulated that the interface between PS I to PS II took place at ca. 1000 BCE. However, Fig 3 demonstrates continuous technological improvement throughout the period of Cluster 2. At the end of TP I, for the first time in the Iron Age, copper efficiency reached the value already attained in LBA (ca. 1.1% Cu), with an even lower STD. In the following Technological Phase (TP II), the copper concentration in the slag dropped (0.38% Cu) and the STD decreased even more significantly to an unprecedented level (0.18% Cu), as clearly shown by Ben Yosef et al. [11].

**Table 1. Comparison between the "technological phases" to the "production systems".**

| Geo-political situation | Sites | Production System (PS) after: Ben-Yosef et al. 2019 | Number of Samples | Mean copper (%) | Standard Deviation (STD) (%) | References | Technological Phases (TP) |
|---|---|---|---|---|---|---|---|
| Egyptian presence (LB IIB) | Timna 3 | PS 0 1300–1140 BCE | 45 | 1.1 | 0.77 | [7,8] | TP I Smelting is based on LBA technological principles LBA–10th century |
| Indigenous Arabah population (Early Iron Age) | Timna & Faynan | PS I 1140–1000 BCE | 56 | 1.65 | 0.87 | [11] | |
| | | PS II 1000–925 BCE | 46 | 1.03 | 0.57 | | TP II New smelting system Mid-late 10th cent.–late 9th |
| | | PS III 925–830 BCE | 41 | 0.51 | 0.18 | | |

The Cluster Analysis shows that a significant decrease in copper content in the slag, attesting to improved production efficiency, took place already at the time of KAJ C1. According to the excavators, KAJ C1 was abandoned in the second half of the 10th century [10, pp. 370–372]. Nevertheless, the wide range of radiocarbon dates does not allow precise dating if this event which may or may not have preceded the Sheshonq I campaign in 925

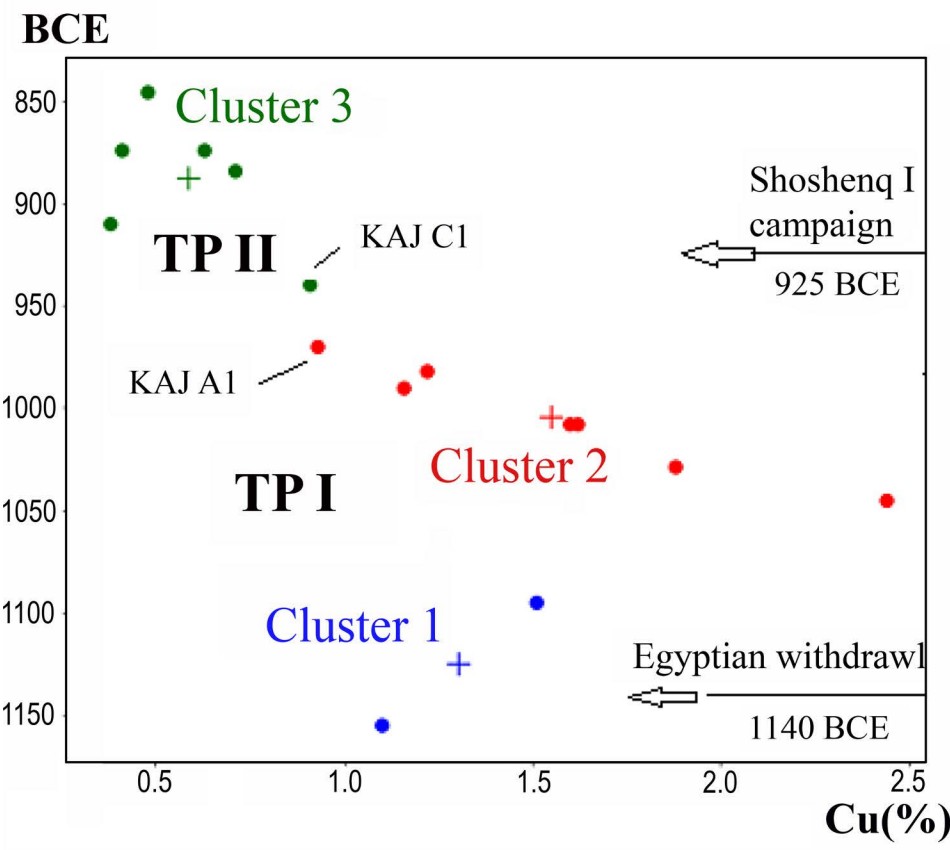

**Fig 3. Results of the cluster analysis.** Cluster 1 shows a decrease in copper-production efficiency after the withdrawal of the Egyptians from Timna. Cluster 2 shows a gradual and continued improvement, which reaches roughly the same values achieved at Timna during the LBA. Cluster 3 shows advanced copper production of TP II. Cluster data information provided in S1 Table.

BCE. However, despite the fact that KAJ C1 was assigned by our analysis to Cluster 3, there are several technical reasons pointing to its affinity to Cluster 2:(i) The copper contents in the slag is considerably higher than the rest of the samples included in this cluster 3, and very similar to that measured in slag from KAJ A1, which is included in Cluster 2. (ii) The STD is considerably higher than that of the rest of the samples in cluster 3, (iii) The improved tool kit of TP II was not identified among the remains in this site. It thus appears that the evidence from KAJ C1 may be taken to reflect a crucial step towards the initial development of TP II: i.e., an unsuccessful attempt to increase efficiency using the same hardware of TP I. This interface between the clusters appears to demonstrate the reason why the local craftsmen decided to develop a new technology, TP II. Such a process is naturally invisible in the archaeological record.

## Punctuated equilibrium or gradual improvement?

Analysis of the grand averages of PS I–PS III, marked by the enlarged triangles in Fig 2, shows a gradual and linear improvement of production efficiency over time (Y = 15X+808, $R^2$ = 0.99). Nevertheless, this simplified calculation fails to satisfy the following requirements: it ignores the adaptation period of Cluster 1, in which production efficiency had declined before it increased, and is also limited to the mean results of each PS, thus, failing to take into consideration the results of each individual site.

Fig 2 in Ben Yosef et al. [11] brings three separate sites (Fig 2A–Khirbet e-Nahas [KEN]; Fig 2B–Timna site 30 [T30]; Fig 2C–Khirbet al-Jariya [21, KAJ]) in support of the proposed paradigm of "punctuated equilibrium" which, as stated above, assumes that there had been a period of relative stasis in the quality of copper production until a critical leap occurred at the beginning of PS III, triggered by the Egyptian campaign. However, two of the figures, Fig 2B and 2C, cannot be used for this analysis: Fig 2B shows results from T30, where only two layers exist, which are not sufficient to show any trend/nature of the development; Fig 2C shows the results from KAJ, where none of the strata post-date the Sheshonq I campaign and thus, these results are incapable of supporting the punctuate equilibrium paradigm. The only relevant stratigraphic evidence brought by Ben-Yosef et al. [11, Fig 2A] is from Area M in KEN. However, the results from KEN show linear behavior ($R^2$ = 0.98) for the entire range of dates (11th–9th centuries), with no signs of "stasis" or "leaps" before or after Sheshonq I's campaign in ca. 925 BCE (see Fig 4).

Summing up, in the data presented in Ben-Yosef el al. [11], there is no any indication for periods of "stasis" or for sudden "leaps", but rather, a gradual and continuous improvement in copper efficiency that is mostly linear in form. Thus, the proposed suggestion to apply the "punctuated equilibrium theory", to explain the developments of copper production in the Arabah seems unsuitable in this case.

## Advanced techniques and managerial capabilities in the Arabah industry

The production of copper requires four basic requirements: (i) adequate raw materials, i.e., ores and flux; (ii) a constant supply of fuel, i.e., charcoal; (iii) a smelting protocol; (iv) the proper equipment to execute the process, particularly bellows, tuyères and furnaces.

It is assumed that for the development and production optimization, the following advanced means were used:

a. **trial and error**: an empirical method in which the final routine conditions are reached through a series of optimisations of the production variables through strict monitoring of the final yield.

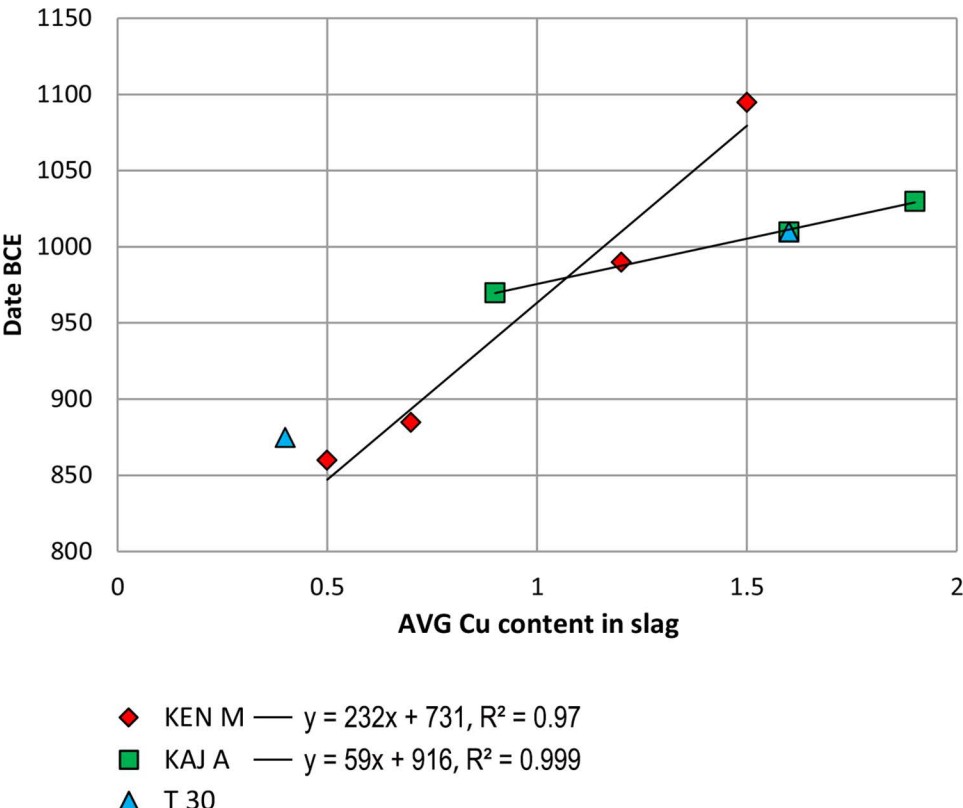

**Fig 4. Linear improvement in copper production at Timna and Faynan.** Based on ref. [11, Fig 2A–2C].

b. **scaling-up**: a developmental technique intended for increasing the size of an existing production component (i.e., tuyère, furnace) according to a predetermined scale by using known production techniques. This procedure is beneficial for reducing experimental costs and developmental risks.

c. **managerial quality:** an efficient management cooperating closely with the technical team, monitoring and advancing the technical developments and day-by-day operation.

This terminology has been coined during the modern era within the discipline of industrial engineering, whereas the Arabah people might simply have referred to it as "common sense", "intuition" or "good engineering and managerial practices". Nevertheless, such conclusions need a further discussion.

Anthropologists have shown that ancient peoples showed "good practices" guided by "common sense", "intelligence", "intuition" and "rationality". Such practices would allow them to accomplish and to maximize their best of interests [22]. Israel Aumann [23], recipient of the Nobel Prize in Economics, assumes that advance achievements could have been reached through a rational intuition he designates as "rules of thumb". Evidence presented by Henshilwood et. al. [24] shows that humans living at least as early as 35,000 years ago had cognitive abilities similar to that of modern humans. Schiffer and Skibo [25] describe behavioral chains of activities by which craftsmen of earlier periods operated: through the integration of technical choices within a process of "trial and error". The set of integrated technical choices that arises from this "trial and error" process is termed by Schiffer and Skibo as "primary technology". This is exactly what

was demonstrate in the Arabah—how small and successive steps could lead to advanced technologies in the metallurgy of copper. It is also be assumed that such an advanced thinking was also demonstrated at the managerial level of the Arabah, as will be discussed later. Warburton [26, p. 170, 173] sums up his view on the advanced ancient Egyptian economic:

> Even the fragmentary evidence from ancient Egypt confirms the interlocking markets where prices resulting from general equilibrium were available. The fact that the copper and silver appearing in these transactions were themselves parts of the international market economy [...] [and] international equilibrium prices [...] confirming the general lines of Keynes's General Theory.

Thus, through clever and practical thinking provided by intuitive "rules of thumb", ancient people–were already able to apply advanced thinking such as the "trial and error" method and the principles of Keynes's General Theory thousands of years ago.

**Trial and error and its assumed implementation in the Arabah industry.** In the attempt to specify the exact technique used by the Arabah craftsmen for monitoring production, a crucial question arises: today, the gradual improvement and efficiency of the production process is detected, indirectly and inversely, by using advanced analytical techniques that measure the residual copper left in the slag; the lower the copper contents, the better and more efficient was the production. However, how was the quality of the smelting procedure monitored 3,000 years ago? In antiquity, production efficiency could have been tested directly by weighing refined copper, using a simple balance scale, well known from the 3$^{th}$ millennium BCE and whose accuracy is most appropriate for this task [27]. The use of a balance scale is depicted, for example, in wall paintings from the Old Kingdom Tomb of Mereruka, Saqqara [28]. Our principle assumption suggests that trial and error could successfully lead to the gradual, step-by-step improvement in smelting by changing one variable in the production process at a time, while keeping all the other variables constant. It is assumed that only one change could be implemented at each step, since otherwise the interpretation of the results would not be possible. For example, should, in one experiment, both the tilting angle *and* the specific location of the tuyère within the furnace be changed, resulting in a 2% improvement in copper yield, it would not be possible to discern between the individual effects of each of the independent variables, and the test would be useless. Hence, presumably, in each trial, the specialists started from the latest working procedure, allowing for a minor and successive change of one variable. This empirical approach was a remarkably long and tedious process, but would eventually lead to significant improvements in copper production.

Naturally, such a development process would not leave detectable evidence, as limited amounts of slag would have been produced during the trial and error experiments in relation to the huge amounts of slag produced during ongoing production. Thus, identifying the "experimental" slag among the "conventional" pieces would be akin to "finding a needle in a haystack".

Significantly, at the end of TP I, when residual copper in slag reached only ca. 1%, it became more and more difficult to achieve additional improvement and drastic measures were needed. Assumingly, this led the local craftsmen to the necessary developments in the hardware of TP II.

**Scaling-up technique and the origin of the TP II tuyère.** Before LBA, copper production in the region was based on wind-operated furnaces, rather than on forced draft using bellows, pipes and tuyères [19,29]. The "short tuyère" used throughout the LBA and the Iron Age I in the Arabah was the same as the tuyère used in Sinai during the LBA [6, p. 29], [10, pp. 702, 989], [30, p. 26], while the long tuyère was introduced in the late 10$^{th}$ century BCE at Timna Site 30 and at Khirbat en-Nahas [6, pp. 36–48], [10, p. 888]. Al-Shorman [29, pp. 115–116] describes

**Table 2. The three types of tuyères used in LBA and iron age.**

| Type & Technological Phase (TP) | Period | Found location & Reference | Length (cm) | Diameter (cm) | Air flow diameter (cm) | Inclination (degrees) |
|---|---|---|---|---|---|---|
| Short tuyère **TP I** | LBA & Early Iron Age until mid-late 10th century BCE[a] | Southern Levant, Egypt and the Arabah [10, p. 989] | 4–9.2 | 5–8 | 2 | 25–40 |
| Long tuyère **TP II** | Mid-late 10th century-late 10th century | Arabah [10, p. 720] | 32 | 13–16.5 | 2.5 | 20–25 |
| Reused tuyère **(A unique Sinai type)** | Iron Age | Bir Nasib, Sinai [29, pp. 115–116] | 10 | 10 | 2 | 20–25 |

[a]Ben-Yosef et al. [11, p. 9] suggest 925 BCE.

a third tuyère type, designated a "reused tuyère"; such tuyères, dated to the Iron Age, were unearthed at Bir Nasib, Sinai. This innovative device has never been found in the Eastern Desert nor in the Arabah, suggesting that this was a local innovation at Bir Nasib. All three tuyère types are described in Table 2.

The main problem with the operation of the short tuyère was its tendency to become clogged, thus preventing the continuation of airflow from entering the furnace [6, pp. 35, 36], [29, pp. 114–115, Fig 4.26]. This inherent problem is a direct result of the considerably large temperature difference acting upon the tuyère. The inner nozzle's tip is exposed to very high temperatures (up to $1350^0$ C) and to the superheated particles of the slag. At the same time, air enters into the chamber at a relatively low temperature (ca. $40^0$ C). Therefore, the air tends to cool the tuyère and its surroundings, causing the superheated particles to solidify, resulting in slag formation in the nozzle surroundings, as demonstrate schematically in Fig 5, ultimately leading to nozzle clogging. Heat flow considerations show that enlarging the dimensions of the tuyère leads to a better temperature isolation of the nozzle, which reduces the likelihood of clogging. For example, increasing the length of the nozzle by 10 mm reduces the temperature gradient by ca. $150^0$ C (estimated from [29, p. 222, Fig 7.15]).

The existence of reused tuyères indicates that the craftsmen at Bir-Nasib, Sinai faced the same clogging problem that existed in the short tuyère in the Arabah. However, they arrived at different solutions: (i) replacing the clogged nozzle with a new one, up to seven times and (ii) increasing the dimensions of the short tuyère, although not to the same large dimensions selected for the "long tuyère" in the Arabah (Table 2).

The small and long tuyères are similar in their principle design and raw materials and had a very similar production technique [10, pp. 702–705]. Both are composed of two primary components: a front part ("nozzle") made of refractory clay, and a back part made of reddish clay, rich in crushed-slag inclusions. Notably, the long tuyères lack finishing marks, such as cloth-wiping, on their exterior, as do the small ones. Instead, in the same location, there are marks of organic material such as grass, or weeds. In addition, the long tuyère required a multi-layered construction because of its increased size [10, pp. 702–705]. It seems probable that the different tuyère prototypes were tested through trial and error for optimization of the design parameters. It is assumed that the same scaling-up technique was also used for increasing the dimensions of the furnace, intended to improve the productivity.

**The role of the managerial quality of the Arabah on the economic success.** It has already been shown that humans living thousands of years ago had cognitive abilities similar to those of modern ones [24], and that their economic perspective conformed with the general lines of Keynes's General Theory [26]. In a similar way, Schiffer and Skibo [25] demonstrate the importance of "trial and error" in the ancient past, while in the modern era, "experimentation"

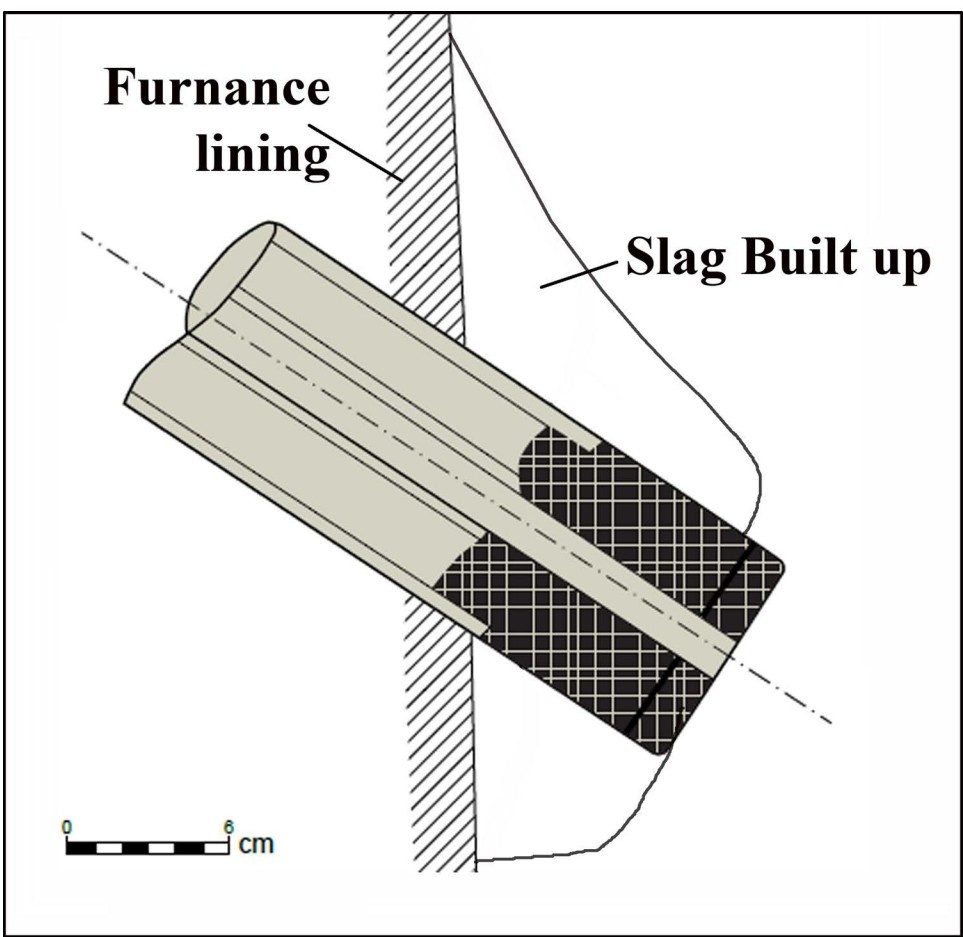

**Fig 5. Schematic representation of a typical tuyère in a clogging process.**

and "trial and error" remains an important and emphasized ideology for more than 400 of the world's largest and most successful firms as shown by Patel and Pavitt [31] using "experimentation" and "trial and error".

Two main questions will be addressed in this discussion: (i) is it possible to recognize the existence of a competent managerial team at Timna and Faynan based on the archaeological evidence? And, (ii) what was the mandate of such a management?

Similarity in paraphernalia and by-products revealed a centralized management that controlled both Faynan and Timna [11]. Naturally, the main objective of the management was to increase copper production. To this end, the managerial team had the mandate to close unproductive mines and smelting sites, to open new more efficient ones and to search for alternative sources, i.e., secondary smelting of existing slag [10,19].

The decision to employ secondary smelting of EBA slag was, most likely, a very difficult choice to make. This would have required considering the following subjects: how does the copper content in the mineral sources compare to the EBA slag? What is the amount of energy (defined by volume of charcoal) needed by each of these alternatives in order to produce one unit of copper? If the assumption is made that the content of copper in a natural mineral is much higher than that left in the EBA slag, the question is raised of whether or not it would be advantageous to save on the cost of mining in return for the additional energy costs required

for smelting the lesser quality slag? It is logical to assume that the management of the copper industry in the Arabah was able not only to raise these questions but also to have a technical team that could help answer them. It is notable that excavation in Timna (T34, "Slave Hill") showed evidence for the existence of high level personnel [32], probably from the technical and the managerial teams, that enjoyed high-quality food sources and fine items of clothing.

An essential task of the management was to develop, by themselves or through proxies, new international markets for the copper. The Arabah succeeded doing so via the King's Highway, which led Moab to establish lines of fortresses along her Eastern border as shown by Homès-Fredericq [33], Finkelstein and Lipschits [34] and Lev-Tov *et al*. [35]. Recent evidence demonstrates that the markets of the Arabah extended as far as Olympia and other Aegean destinations [36].

Summing up: The main tasks and the mandate of the managerial level of the Arabah were: (1) day-by-day operation, (2) expanding production and improving efficiency in Timna and Faynan, (3) closing unproductive mines and smelting sites and opening more efficient ones, (4) finding alternative sources like secondary smelting, (5) developing international markets.

It would be very difficult to accept the possibility of an Egyptian involvement in the managerial level during the period between Early to Late Iron Age because such presence would have required the physical presence of qualified Egyptian personnel in the Arabah. However, such evidence has not been found.

## Discussion

### Regional copper production

Most of the operating sites in the Sinai can be dated to the Early Bronze, Middle Bronze and Late Bronze Ages, while only a few can be dated within the Iron Age [30,37,38]. Rothenberg [6] reported that after the 15th century BCE, the Egyptians failed to continue the operation of their largest copper-production center near Bir Nasib, Sinai, however, it was operated during part of the Iron Age and later during the Nabataean, Roman and Byzantine periods [30,39]. Additional copper-smelting sites were operated in the northern part of the Eastern Desert of Egypt. These sites are dated to the Late Prehistoric, Old Kingdom and Ptolemaic/Roman periods in Egypt [30, p. 50]. Smaller copper production sites were also located at Qurayyah and Tayma in the Arabian Peninsula. The Qurayyah site was active during the Late Bronze Age, while that of Tayma was dated to the Roman/Late Roman periods; neither of them were operational during the Iron Age [40]. However, it must be noted that the archaeology of northwestern Arabia is only in its infancy and very little is known about its copper ores [41].

Throughout the LBA, in addition to Egyptian-local production, copper was imported from Cyprus, which was one of the richest copper sources during this time [18]. At the end of the LBA, Egypt withdrew from many of the smelting regions and was also affected by the collapse of the Mediterranean copper trade. Kassianidou [42, p. 267] assumes that after the trade collapsed, the Cypriots turned their interest to the Aegean and beyond to the Central Mediterranean, where they found new markets. In addition, the process of iron dissemination to Egypt was slower than in the Levant and reached its peak only in the second half of the first millennium BCE [43, pp. 167–168]. Therefore, the need for copper consumption inside Egypt during the Iron Age did not decrease, as it had in other areas throughout the Levant where copper was replaced by iron. All these factors most probably led to a copper deficiency in Egypt during the first millennium BCE [44]. Under these circumstances, it would be expected that Egypt would increase its internal copper production, but just the opposite happened. Furthermore, since copper production in Egypt increased considerably in later periods, such as the Nabataean, Roman and Byzantine periods [30, p. 50; 49, p. 17–18], the deficiency in

Egyptian copper production during the Iron Age cannot be related to the exhaustion of local mineral sources. Alternatively, poor managerial capabilities and lack of technical skills may have prevented the Egyptians from exploiting inner copper resources and fulfilling their internal demand.

Thus, the macro historical circumstances support the assumption that after the withdrawal of the Egyptians from Canaan, the indigenous people of the Arabah had no other alternative rather than to take the initiative to develop their own technical and managerial capabilities, benefitting from the advantage of having few copper competitors, if any, in the arena. Thus, I agree with Fantalkin and Finkelstein [44] who argued that the objective of Sheshonq I's campaign was to "preserve and promote" the copper production in the Arabah, as, at that time, the Arabah source "must have been the major—if not only- source of copper for Egypt". As Sheshonq I was aware of the huge reduction in copper production in Egypt during the Iron Age [30,37,38] he decided to secure a steady copper supply from the Arabah to Egypt.

## Egyptian involvement

In general, the policy of Egypt in regards to external production sites ranged from indirect influence to direct involvement in the production and/or the organization of production [1,16]. Petrie [45, pp. 109–110] describes some of the Egyptian procedures with the locals, such as dividing the labour into numerous different tasks and camps for "obtaining great results from small minds". The recent publication by Ben Yosef et al. [11] describes the Egyptian's doctrine as "imposed or triggered". However, if true, this would require at least some degree of their physical presence at the copper-production sites, which is not supported by the archaeological evidence at hand.

A scarab bearing the name of Sheshonq I [12], and other Egyptian artefacts, were reported from Faynan. These include glass beads, small figurines and seals as well as 10 Egyptian amulets (mostly dated from the reigns of Siamun and Sheshonq I), covering most of the 10th century BCE (ca. 980 to 900 BCE) [46, p. 759]. However, none of these artifacts can serve as solid proof of Egyptian physical presence at Faynan and may have reached the region as a result of trade. Similar finds from other sites in the southern Levant were not taken as proof for Egyptian presence (e.g., numerous Egyptian amulets unearthed at Tel Rehov [17]. The archaeological evidence for the physical presence of Egyptians in the southern Levant is well known from the time of the 19th and 20th Dynasties, when at sites such as Jaffa and Beth Shean, such presence was reflected in the architecture, inscriptions, sculptures, cult and the robust occurrence of Egyptian and Egyptianizing pottery, all of which were present at Timna during LBA, but completely lacking at Faynan (e.g. [47,48]).

Concerning the technological and managerial aspects of Egyptian involvement in the period between Early and Late Iron Age, It seems difficult to accept the paradigm that the advanced TP II technology was introduced by the Egyptians following the military campaign of Sheshonq I, for the following reasons:

(i) TP II technology was never identified in Egypt or Sinai, therefore, most probably, could not have been introduced by the Egyptians.

(ii) If TP II was an Egyptian technology, it would have required the physical presence of qualified Egyptian personnel in the Arabah. However, as noted above, no substantial evidence [49] for such a presence can be gleaned from the finds.

## Summary and conclusions

The prevalent explanation for the processes that took place following the Egyptian departure from the Arabah in the mid-12th century BCE suggests that for most of this period there was

a continuation of LB technology with a small degree of local development. This lasts until an external event, the Sheshonq I campaign to Canaan in ca. 925 BCE, introduced innovated technologies and a new organization, thus resulting in a major qualitative change in the copper smelting technology. This was suggested to have taken place according to the model of "punctuated equilibrium", which emphasizes a dramatic change (or 'leap') following periods of stasis [11].

In contrast, this work suggests that the gradual and continuous developments in copper technology in the Arabah during the course of the Early Iron Age, following the departure of the Egyptians in ca. 1140 BCE and up to the abandonment of the site in the late 9th century, were attained independently through ongoing investment and innovation by the local crafts-men. This process was accomplished by the continuous application of techniques that can be defined, according to modern industrial engineering terminology, as "trial and error" and "scaling-up". It is assumed that sometime during the 10th century BCE, when the local crafts-men achieved the peak quality attainable by the original Egyptian technology (understood by the residual ca. 1% copper in the slag), they decided to enhance the basic hardware, TP I, and set forth development on a new tool kit (mostly new tuyères and large furnaces), defined here as TP II. This developmental stage left no archaeological evidence. After the developmental stage was successfully terminated, sometime in the middle of the 10th century BCE, the large tuyère and enhanced furnace were introduced simultaneously at the smelting sites in a "sud-den event".

On the other hand, it is not possible to find any tangible evidence for the existing parallel paradigm, which suggests that such changes and other improvements were the result of extra-neous or sudden intervention by foreign powers.

I accept the alternative explanation of Fantalkin and Finkelstien [44] regarding the impact of the Sheshonq I campaign on the Arabah copper industry; namely, that the Egyptian endeavor was, in fact, the result of an existing copper shortage in Egypt at that time. Thus, the technological advances in the Arabah industry during the course of the 10th century BCE were most likely implemented for the sake of coping with the growing demands for copper in the Egyptian market, rendering the Egyptians the consumers rather than the producers and innovating technologists.

## Supporting information

**S1 Table.**
(XLSX)

## Acknowledgments

I would like to thank Naama Yahalom-Mack for assisting me in writing this paper, Nava Panitz-Cohen for her editing and valuable comments, Ruhama Bonfil compiled Figs 3 and 4, and Itamar Ben Ezra illustrated Fig 1. Oded Luria kindly assisted with the statistics.

## Author contributions

**Conceptualization:** David Luria.

**Methodology:** David Luria.

**Writing – original draft:** David Luria.

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
