## [Decision Letter · Decision Letter 0]

28 Sep 2020

PONE-D-20-15801

Copper technology in the Arabah during the Iron Age: Punctuated equilibrium by extraneous intervention or gradual improvement by local craftsmen?

PLOS ONE

Thank you for submitting your manuscript to PLOS ONE. After careful consideration, we feel that it has merit but does not fully meet PLOS ONE’s publication criteria as it currently stands. Therefore, we invite you to submit a revised version of the manuscript that addresses the points raised during the review process.

In particular, reviewers raised concerns with both the methods and framing of the paper. Additionally, there was concern that some conclusions may not be fully supported by the results, and that some interpretations may not be fully grounded in the existing literature.

We look forward to receiving your revised manuscript.

Kind regards,

Bradford Dubik

Academic Editor

PLOS ONE

Journal Requirements:

Additional Editor Comments (if provided):

Reviewers' comments:

Reviewer's Responses to Questions

**Comments to the Author**

1. Is the manuscript technically sound, and do the data support the conclusions?

Reviewer #1: Partly

Reviewer #2: Yes

Reviewer #3: Yes

2. Has the statistical analysis been performed appropriately and rigorously?

Reviewer #1: Yes

Reviewer #2: N/A

Reviewer #3: I Don't Know

3. Have the authors made all data underlying the findings in their manuscript fully available?

Reviewer #1: Yes

Reviewer #2: Yes

Reviewer #3: No

4. Is the manuscript presented in an intelligible fashion and written in standard English?

Reviewer #1: Yes

Reviewer #2: Yes

Reviewer #3: Yes

5. Review Comments to the Author

Reviewer #1: General Comment: This is a response to a bad paper published in PLoS One in 2019. While I agree with your substantive criticisms of that paper, I am not recommending that PLoS One accept your response. I have two reasons for this. The first is that both the original paper and your response are not of sufficiently wide interest for publication in a leading general science journal – they belong in either a regional archaeological journal, or in Journal of Archaeological Science: Reports. My second criticism is that many of your archaeological conclusions seem to me to be speculations that bear no necessary relationship to the evidence or analysis actually presented in the paper.

Detailed comments:

Lines 35-41. This is not an introduction! You need to provided context. Where is the Arabah? Why should anyone be interested in this topic? Why is Feynan significant in the history of metallurgy? Why was there a copper industry in such a remote area? What is the chronological and geological relationship between Timna and Feynan? Provide citations to prior research here.

Line 57. There were no geologists, mining engineers or physicists in the LBA! This is reading the present back into the past - what historians call the “presentist fallacy”.

Lines 60-62. We actually do know something about Egyptian mining technology (from gold mining in the Eastern Desert) that is contemporary with the Egyptian presence in the Arabah! You should read about it and compare with the technology at Feynan.

132-133. It would be helpful to have a table of amounts of slag in each period so that the reader doesn’t have to go to the study cited. These changes in output are really important to your argument.

152-153. I agree with your criticism here – arguments about technological innovation should be based solely upon the evidence from studies of the technology.

174-176. This is a valid criticism. The ores at Feynan are not identical to those at Timna, so there would clearly have been some experimentation at the beginning of their exploitation.

Fig. 5. Correct label (“furnance”).

230-235. You ignore here the evidence that they present in their Figure 2D and their Figure 4, which suggests to me that the “leap” in efficiency may simply be a consequence of the discovery that using manganese rather than iron oxides greatly improved the fluidity of the slag. This cannot be attributed to superior Egyptian technology – it is simply a consequence of the fact that some Faynan ores contain manganese. Discuss this with respect to your periods TPI and TPII.

236-253. This is another instance of the “presentist fallacy”. It is completely inappropriate to apply concepts from modern business schools to production more than 3000 years ago! A better place to look for suitable ways of thinking about this in ethnographic and historical studies of indigenous African metallurgy. If you look for example at the work of Philip de Barros at Bassar in Togo, or of the Swiss group (Eric Huysecom, Vincent Serneels, etc.) among Dogon ironworkers in Mali, you will find that “scaling up” requires neither the involvement of a state, nor even a managerial elite. (You can find these easily through Google).

254-283. There is a much simpler way to monitor copper losses, which is to look at freshly fractured surfaces of copper slag under low magnification (10x-20x). The size of metallic copper prills retained in slag is closely proportional to viscosity of the liquid slag; the smaller the prills, the better the recovery.

Table 2. Can I suggest that you use “short tuyere” instead of “small tuyere” and “long tuyere” instead of “large tuyere”? It will make your discussion much easier to understand.

Lines 331-355. This section is absurd, and another example of the “presentist fallacy” to which I referred above. How can the findings of a study of modern (late capitalist) management possibly be relevant to the management of copper smelting in the Bronze and Iron Ages?

342-344. This is not evidence of “management”! Techniques can, and do spread, by imitation – they do not have to be imposed by “management”. You provide no evidence whatever of the existence of “management” at this time.

392-394. “Furthermore, since copper production in Egypt increased considerably in later periods, such as the Nabataean, Roman and Byzantine periods, the deficiency in Egyptian copper production during the Iron Age cannot be related to the exhaustion of copper mineral sources.” I don’t know of any evidence for such an increase in Egypt – and you provide no citations. The main problem with metal production in Egypt has always been lack of fuel, not scarcity of ore. This is why the Egyptians imported copper from the Sinai, the Arabah and Cyprus, and iron from Nubia.

404-405. This is wild speculation on your part. We know essentially NOTHING about Egyptian copper smelting technology around 1000BC except for a couple of depictions of bellows in use on carved or painted panels.

422-432. This is your best counter-argument. One would expect to find material evidence for an Egyptian presence, in the form of buildings, shrines, etc. – as with the earlier Egyptian presence.

476-478. “It was a combination of innovative individuals, excellent local managerial quality and emerging market demands that dictated the impressive surge and success of the technology and organization of the Arabah industry at this time.” This conclusion is unsupported by any evidence presented in the paper. Innovative individuals? Yes, that seems likely. Excellent local managerial quality? No evidence at all. Emerging markets? Not even mentioned before this sentence!

In summary, I think that you effectively destroy the arguments of Ben-Yosef et al. for a revolution in production because of the introduction of new technology by the Egyptian state. You do not however provide any evidence in support of your anachronistic claims for modern systems of management at Faynan during the Iron Age. I think that this article should be revised and then sent to a regional archaeological journal.

Reviewer #2: First of all, congratulations for the well-structured and well written paper. It does not happen very often to be able to accept a paper for publication with no modifications.

The text flows smoothly and all your theories and conclusions are properly supported by a sound and well-argued reasoning, as well as by updated and relevant bibliography.

One small remark: In Figure 1, it would be good to have both a general map of the region and the map of the area with the sites mentioned in the text (that you already have). This would allow a reader who is not fully familiar with the region to better understand how it is situated with respect to Egypt and the Arabian Peninsula.

And one personal thought: I feel that this type of articles focused on “counterstriking” someone else’s theory can be perceived in a very sensitive way. Although I do not believe that your writing style is offensive at all, sometimes reading through the text again (especially the first part) may help to smoothen the tones even more and giving the reader the clear feeling that the paper is just proposing an alternative theory, and that it does not want to be in any way an attack to other authors.

Congratulations again for the great work!

Reviewer #3: The manuscript is effectively a response and rebuttal to a 2019 PLoSONE paper, by Ben-Yosef et al., “Ancient technology and punctuated change: Detecting the emergence of the Edomite Kingdom in the Southern Levant.” Briefly, the authors of the present manuscript argue that there is insufficient evidence for an abrupt technological shift precipitated by the arrival of Egyptians.

I am generally in agreement that there are significant issues with the Ben-Yosef et al argument, certainly from the perspective of metallurgical technology, and also perhaps from the perspective of other evidence. It is productive to these discussions to see some pushback from among those working on archaeometallurgy in the Southern Levant. Given that the original paper was published in PLoSONE, it makes sense that the rebuttal should be published in the same venue to aid discoverability.

However, I think that the critique could be better formulated. There are methodological and framing issues with some of the critiques, others seem less relevant, while other avenues of critique are unexplored. Below, I note several areas for improvement.

150-160. This framing of the critique, claiming that only “technological” evidence should be used in creating models of production system, not “archaeological and historical considerations,” is ineffective. The point seems to be that the focus should only be on analysis of direct production residues—slags, tuyères, and the like.

Most scholars of technology would agree that technological systems consist of much more than just the immediate techniques and behavioral sequences involving the act of production. In discussing organizational and management aspects, the authors themselves seem to acknowledge this. Thus, it seems problematic to claim that archaeological and historical considerations shouldn’t be included, with the goal of producing a “purely technological viewpoint.” Broader social, political, and economic considerations, revealed through analysis of the broader archaeological and historical evidence, are very much an important part of analyzing a technological system.

I suspect the authors realize all this, but the framing here could be improved.

It seems the multivariate statistical techniques (161ff) are being done on averages, rather than on the primary data, which is available as supplementary information alongside the Ben-Yosef et al paper. This is potentially problematic. Aside from a justification of the use of averages, I’d also want to see some methodological discussion about applying the cluster analysis to a dataset with a combination of continuous chemical data (Cu%) and radiocarbon data, which are calibrated date ranges with probability density distributions that are not normal.

The Ben-Yosef et al. 2019 figure has its own problems (the use of only 1σ error bars for the date ranges, when 2σ is pretty much standard for reporting in archaeology).

Given these potential issues, I’m not sure that the statistical analysis is helpful or necessary to build the critique. What can and should be stated (without any need for statistical analysis), is that there is really no abrupt shift in the copper content of the slags as shown in the Figure 2 of this paper (Figure 3 of Ben-Yosef et al.). Lines 208ff makes this point quite effectively without recourse to cluster analysis. What you see is really a gradual linear trend, one that would likely appear even more gradual if the individual analyses were plotted rather than just averages. Visualizing this by recreating that figure with raw data rather than averages would actually quite useful, and would probably help the authors state their case. The data are available as supplementary information on the Ben-Yosef et al 2019 paper, so it should not be difficult to do.

Furthermore, a critique of the Ben-Yosef et al. paper would do well to point out some potential limitations on the use of Cu content as a proxy for smelting efficiency.

--First, slags from pre-modern smelting slags are often quite heterogeneous. Because copper is heavier than the silicate matrix, it will tend to sink, so samples of slag taken from the top of a furnace slag cake will have less copper than a sample taken at the bottom. Many of these differences will be large compared to the average differences measured chronologically (a total range of about 1.5 wt% for the averages shown in figure 2.) For similar reasons, tap slags may differ in consistent ways from furnace slags. It may be worth looking into the Ben-Yosef et al 2019 supplementary data and methods information to see whether they took this into consideration. At the very least, it would be nice to have reassurance that they compared like to like (e.g. tap slags to tap slags).

--Second, slag composition is dependent significantly on the composition of the ore. Ben-Yosef et al mention this briefly, but it is worth discussing more fully. If the copper content of the ore used declined over time, which one might expect as individual mines are worked out, the copper content of the slags would also decease. Could the decreasing copper content of the slags be explained by the progressive working out of the mines exploited during the Iron Age? This is probably worth considering.

Discussion of these issues would strengthen the critique.

260-273. This discussion is not well supported. We have very little idea whether ancient smelters would have approached invention and experimentation in the same controlled way that R&D firms do today, only modifying one element of a procedure at a time. It is entirely possible that improvements were the result of post-hoc modification upon noticing an improved yield after an accidental modification of established procedure (i.e. copy errors, to use the term from evolutionary theory), rather than a goal-oriented, controlled experimental testing.

278-280. This speculation about the copper-iron chunks goes too a bit too far. These copper iron chunks are referenced periodically, but they have not been studied well enough, especially with respect to their microstructure, to be able to say what process they come from.

In the section “Advanced Techniques and Managerial capabilities in the Arabah industry” the authors do not do a good job of explaining how these processes (trial and error, scaling up, administration) differ from those proposed in the 2019 paper. After all, the 2019 paper does argue for a gradual improvement in Cu smelting technologies prior to the alleged sharp break corresponding with Sheshonq’s arrival. Trial and error, and scaling up can all be incorporated into the evolutionary model proposed by Ben-Yosef et al. It’s the “punctuated” part of the “punctuated equilibrium” concept where the models differ. In particular, the sub-section discussing the importance of managerial quality doesn’t help us distinguish whether the relevant administrative team was of local extraction or foreign.

380-381. This final sentence is important. Given the adjustments to the Timna chronologies over the last 15 years, how accurate are the chronological designations for the Sinai copper exploitations, particularly LBA/EIA?

389-390. It doesn’t quite follow how the slower adoption of iron in Egypt contributed to the copper deficiency. Further elaboration needed.

404-405. What is the evidence that copper technologies in the Arabah were superior to those in Egypt? Missing here is a discussion of local Egyptian copper smelting technologies for comparison.

437-438. Similarly: could it be that TPII technologies haven’t been found in Egypt or Sinai because the Arabah has been the subject of far more intensive and extensive archaeological research? Absence of evidence isn’t evidence of absence, especially given the disparities in archaeological research.

440-441. “No evidence for such a presence can be gleaned from the finds.” I would rephrase this, because proponents of an Egyptian presence could counter that the presence of a few overseers or a small military detachment might not leave a massive archaeological signature, and there are a handful of Egyptian objects as noted above. What the authors could reasonable say is that evidence for a substantial Egyptian presence is limited or equivocal.

Figure 5. I think “Slag buildup” is what is intended, not “slag built up.” And it’s Furnace not Furnance.

Overall, I think that once these revisions have been made, this critique will form a useful contribution to the literature on technological change.

6. PLOS authors have the option to publish the peer review history of their article (what does this mean? ). If published, this will include your full peer review and any attached files.

**Do you want your identity to be public for this peer review?** For information about this choice, including consent withdrawal, please see our Privacy Policy .

Reviewer #1: **Yes:** David Killick

Reviewer #2: No

Reviewer #3: No

---

## [Author Response · Author response to Decision Letter 0]

15 Jan 2021

Response to the Reviewers

The new title:

Copper Technology in the Arabah during the Iron Age and the Role of the Indigenous Population in the Industry

The change in the title is a direct result of a remark received from the second reviewer, as discussed in the following.

The original title deleted:

Copper technology in the Arabah during the Iron Age: Punctuated equilibrium by extraneous intervention or gradual improvement by local craftsmen?

Reviewer #1:

General Comment: This is a response to a bad paper published in PLoS One in 2019. While I agree with your substantive criticisms of that paper, I am not recommending that PLoS One accept your response. I have two reasons for this. The first is that both the original paper and your response are not of sufficiently wide interest for publication in a leading general science journal – they belong in either a regional archaeological journal, or in Journal of Archaeological Science: Reports. My second criticism is that many of your archaeological conclusions seem to me to be speculations that bear no necessary relationship to the evidence or analysis actually presented in the paper.

Faynan and Timna were some of the largest copper production facilities excavated in the ancient world. They have the potential to shed light on important questions such as the mode of operation in such industrial venues, how technology changed and how this complex industry was run successfully. Contrary to the reviewer comments I believe that such questions are of wide interest to a diverse range of scholars.

Among these important issues discussed here are: (i) the role of the managerial team in Arabah, (ii) the use of cluster analysis in identification of the complex stages of the technical development along LBA-Iron Age, (iii) a detailed technological explanation of why the new technical development of TP II was mandatory, (iv) presenting a different view regarding the important issue of the scarab bearing the name of Sheshonq I,(v) a reconstruction of the main development procedures used in the Arabah, the "trial and error", and running operation, (vi) introducing the importance of the "scaling up" procedure, (vii) criticizing the paradigm of "punctuated equilibrium", (viii) the conclusion that the advances in the industry were achieved by the locals.

An utmost effort was made in this revised version to provide all the evidence needed for each of the issues discussed.

In addition, I completely share the third reviewer’s opinion saying: "it makes sense that the rebuttal should be published in the same venue to aid discoverability."

Detailed comments:

Lines 35-41. This is not an introduction! You need to provided context. Where is the Arabah? Why should anyone be interested in this topic? Why is Feynan significant in the history of metallurgy? Why was there a copper industry in such a remote area? What is the chronological and geological relationship between Timna and Feynan? Provide citations to prior research here.

The reviewer is right.

The "Introduction and Background" was revised; see lines 34-80 in the new version.

Line 57. There were no geologists, mining engineers or physicists in the LBA! This is reading the present back into the past - what historians call the “presentist fallacy”.

The above statement was made by Avner el al., (2014). I don’t accept it but it was important to quote this paradigm. Unfortunately, I ignored writing the reference of Avner el al., (2014) and the reviewer had the reasons to think that this quotation is mine.

Please see line 73 in the new version:

Avner et al. [5] also suggested that the indigenous people were the geologists, the mining engineers and the physicists behind this industry.

Lines 60-62. We actually do know something about Egyptian mining technology (from gold mining in the Eastern Desert) that is contemporary with the Egyptian presence in the Arabah! You should read about it and compare with the technology at Feynan.

My paper is focused on advanced, high temperature technology of copper and not on mining technology. As gold is found in nature in its final metallic form without a need for "production" it was not discussed here.

132-133. It would be helpful to have a table of amounts of slag in each period so that the reader doesn’t have to go to the study cited. These changes in output are really important to your argument.

The reviewer is right.

See line 43-46:

Following the Egyptian withdrawal from Canaan in the 12th century BCE, local copper production in the Arabah not only continued into the Early Iron Age (11th–9th centuries BCE), but also expanded to include the site of Faynan, reaching an unprecedented scale, particularly at the latter sit: the quantity of slag produced during 1200-1150 BCE was only ca. 1.600 tons, this was gradually increased during 1100 - 1050 BCE into ca. 5.600 tons and in 1000- 950 BCE to ca. 15.600 tons. The peak of production was achieved during 900-850BCE, with ca. 23.000 tons

152-153.

I agree with your criticism here – arguments about technological innovation should be based solely upon the evidence from studies of the technology.

No change is needed

174-176. This is a valid criticism. The ores at Feynan are not identical to those at Timna, so there would clearly have been some experimentation at the beginning of their exploitation.

No change is needed

Fig. 5. Correct label (“furnance”).

It was corrected

230-235.

You ignore here the evidence that they present in their Figure 2D and their Figure 4, which suggests to me that the “leap” in efficiency may simply be a consequence of the discovery that using manganese rather than iron oxides greatly improved the fluidity of the slag. This cannot be attributed to superior Egyptian technology – it is simply a consequence of the fact that some Faynan ores contain manganese. Discuss this with respect to your periods TPI and TPII.

Manganese flux, being an inherent part of the ore, was used in Faynan throughout the period. While it is true that a change in flux, introduced to Timna in the final phase, may explain the improvement, it cannot explain the similar improvement that occurred at Faynan.

236-253.

This is another instance of the “presentist fallacy”. It is completely inappropriate to apply concepts from modern business schools to production more than 3000 years ago! A better place to look for suitable ways of thinking about this in ethnographic and historical studies of indigenous African metallurgy. If you look for example at the work of Philip de Barros at Bassar in Togo, or of the Swiss group (Eric Huysecom, Vincent Serneels, etc.) among Dogon ironworkers in Mali, you will find that “scaling up” requires neither the involvement of a state, nor even a managerial elite. (You can find these easily through Google).

While Barros et al. (2020) show convincingly a change in iron technology in Togo between the Early and Late Iron Age, they do not discuss the origin of the later technology, which was introduced after a gap. In the present study the subject discussed was gradual changes and improvements in an existing technology.

Study of iron production in Mali revolved mainly around the question of how small-scale domestic iron production, utilizing traditional low-tech methods co-existed with (and without being influenced by) large-scale iron production using advanced technology. This is not the case in the Arabah.

Nevertheless, I was convinced that this subject needs more clarifications:

See lines 241 - 267:

This terminology has been coined during the modern era within the discipline of industrial engineering, whereas the Arabah people might simply have referred to it as "common sense", "intuition" or "good engineering and managerial practices". Nevertheless, such conclusions need a further discussion.

Anthropologists have shown that ancient peoples showed "good practices" guided by "common sense", "intelligence", "intuition" and "rationality". Such practices would allow them to accomplish and to maximize their best of interests [19]. Israel Aumann [20], recipient of the Nobel Prize in Economics, assumes that advance achievements could have been reached through a rational intuition he designates as "rules of thumb". Evidence presented by Henshilwood et al. [21] shows that humans living at least as early as 35,000 years ago had cognitive abilities similar to that of modern humans. Schiffer and Skibo [22] describe behavioral chains of activities by which craftsmen of earlier periods operated: through the integration of technical choices within a process of "trial and error". The set of integrated technical choices that arises from this "trial and error" process is termed by Schiffer and Skibo as "primary technology". This is exactly what was demonstrate in the Arabah - how small and successive steps could lead to advanced technologies in the metallurgy of copper. It is also be assumed that such an advanced thinking was also demonstrated at the managerial level of the Arabah, as will be discussed later. Warburton [23, pp. 170, 173] sums up his view on the advanced ancient Egyptian economic:

Even the fragmentary evidence from ancient Egypt confirms the interlocking markets where prices resulting from general equilibrium were available. The fact that the copper and silver appearing in these transactions were themselves parts of the international market economy […] [and] international equilibrium prices […] confirming the general lines of Keynes's General Theory.

Thus, through clever and practical thinking provided by intuitive “rules of thumb”, ancient people were already able to apply advanced thinking such as the "trial and error" method and the principles of Keynes's General Theory thousands of years ago.

Therefore, I would better regard what is called by this reviewer as a “presentist fallacy” into "intuition" or “rules of thumb” that are common to both modern and ancient human logical thinking.

254-283.

There is a much simpler way to monitor copper losses, which is to look at freshly fractured surfaces of copper slag under low magnification (10x-20x). The size of metallic copper prills retained in slag is closely proportional to viscosity of the liquid slag; the smaller the prills, the better the recovery.

This seems to be a valuable method to test in details in the future. Unfortunately, at this time, the only results available are residual copper in the slag.

No change is needed

Table 2.

Can I suggest that you use “short tuyere” instead of “small tuyere” and “long tuyere” instead of “large tuyere”? It will make your discussion much easier to understand.

This remark was accepted and changed in the whole document.

Lines 331-355.

This section is absurd, and another example of the “presentist fallacy” to which I referred above. How can the findings of a study of modern (late capitalist) management possibly be relevant to the management of copper smelting in the Bronze and Iron Ages?

Again, I don’t accept your term “presentist fallacy”.

1. Please see: the correction started in lines 241 – 267 above.

2. Please see the beginning of the "The Role of the Managerial Quality of the Arabah on the Economic Success" in lines 347 - 353.

It has already been shown that humans living thousands of years ago had cognitive abilities similar to those of modern ones [21], and that their economic perspective conformed with the general lines of Keynes's General Theory [23]. In a similar way, Schiffer and Skibo [22] demonstrate the importance of "trial and error" in the ancient past, while in the modern era, "experimentation" and "trial and error" remains an important and emphasized ideology for more than 400 of the world's largest and most successful firms as shown by Patel and Pavitt [28] using "experimentation" and "trial and error".

342-344.

This is not evidence of “management”! Techniques can, and do spread, by imitation – they do not have to be imposed by “management”. You provide no evidence whatever of the existence of “management” at this time.

It is true that ideas do spread by imitation.

Nevertheless, here we deal with a completely different issue: who was responsible to the day-by-day operation, monitoring and controlling of the huge and the complex copper industry?

Namely: how much ore, flux, water, food, trees and donkeys were necessary to carry out operations day-to-day, week-to-week? Is the number of miners sufficient to carry out the work or shall new ones be recruited? and so on.

For such activities imitation cannot help but rather should be decided and imposed by intelligent managers devoted to these tasks.

Please see lines 380 – 363 for the management's responsibilities:

Summing up: The main tasks and the mandate of the managerial level of the Arabah were: (1) day-by-day operation, (2) expanding production and improving efficiency in Timna and Faynan, (3) closing unproductive mines and smelting sites and opening more efficient ones, (4) finding alternative sources like secondary smelting, (5) developing international markets.

Direct archaeological evidence was given to each of these subjects in the text.

392-394.

I don’t know of any evidence for such an increase in Egypt – and you provide no citations. The main problem with metal production in Egypt has always been lack of fuel, not scarcity of ore. This is why the Egyptians imported copper from the Sinai, the Arabah and Cyprus, and iron from Nubia.

The citation was added.

See: line 396

Furthermore, since copper production in Egypt increased considerably in later periods, such as the Nabataean, Roman and Byzantine periods [27, 49],

This relates to: Abdel-Mutelib et al. 2012: 50; Rothenberg 1970: 17–18

404-405.

This is wild speculation on your part. We know essentially NOTHING about Egyptian copper smelting technology around 1000BC except for a couple of depictions of bellows in use on carved or painted panels.

The reviewer is right.

The attempts to compare slag results from the Arabah with the slag results from Sinai and the Northern Eastern Desert were found to be very difficult as only few results are reported at present.

The results found (n=16) were shown by Abdel-Motelib et al. (2012: 37, 40). The average residual copper was 18.8% (!) and the variation between the results was enormous.

As I didn't find more supporting evidence I decided not to publish these results.

Therefore, it is justified to change the wording in this case.

See lines 424 - 429:

I agree with Fantalkin and Finkelstein [39] who argued that the objective of Sheshonq I’s campaign was to "preserve and promote" the copper production in the Arabah, as, at that time, the Arabah source "must have been the major - if not only- source of copper for Egypt". As Sheshonq I was aware of the huge reduction in copper production in Egypt during the Iron Age [33; 34; 27] he decided to secure a steady copper supply from the Arabah to Egypt.

422-432.

This is your best counter-argument. One would expect to find material evidence for an Egyptian presence, in the form of buildings, shrines, etc. – as with the earlier Egyptian presence.

No change is needed

476-478.

“It was a combination of innovative individuals, excellent local managerial quality and emerging market demands that dictated the impressive surge and success of the technology and organization of the Arabah industry at this time.”

This conclusion is unsupported by any evidence presented in the paper. Innovative individuals? Yes, that seems likely. Excellent local managerial quality? No evidence at all. Emerging markets? Not even mentioned before this sentence!

The remark was accepted and the sentence was deleted.

It was a combination of innovative individuals, excellent local managerial quality and emerging market demands that dictated the impressive surge and success of the technology and organization of the Arabah industry at this time.

In summary, I think that you effectively destroy the arguments of Ben-Yosef et al. for a revolution in production because of the introduction of new technology by the Egyptian state. You do not however provide any evidence in support of your anachronistic claims for modern systems of management at Faynan during the Iron Age. I think that this article should be revised and then sent to a regional archaeological journal.

The needed changes were made.

Reviewer #2:

First of all, congratulations for the well-structured and well written paper. It does not happen very often to be able to accept a paper for publication with no modifications.

The text flows smoothly and all your theories and conclusions are properly supported by a sound and well-argued reasoning, as well as by updated and relevant bibliography.

One small remark: In Figure 1, it would be good to have both a general map of the region and the map of the area with the sites mentioned in the text (that you already have). This would allow a reader who is not fully familiar with the region to better understand how it is situated with respect to Egypt and the Arabian Peninsula.

FIG. 1 was corrected

And one personal thought: I feel that this type of articles focused on “counterstriking” someone else’s theory can be perceived in a very sensitive way. Although I do not believe that your writing style is offensive at all, sometimes reading through the text again (especially the first part) may help to smoothen the tones even more and giving the reader the clear feeling that the paper is just proposing an alternative theory, and that it does not want to be in any way an attack to other authors.

Congratulations again for the great work!

I fully accept the recommendation of the second reader to "smoothen the tones even more". To this end the title of the paper was changed to a less offensive wording. Doing so will signal to the readers that the paper is focused on technology and managerial issues rather than “counterstriking” Ben-Yosef et. al.,

Reviewer #3:

The manuscript is effectively a response and rebuttal to a 2019 PLoSONE paper, by Ben-Yosef et al., “Ancient technology and punctuated change: Detecting the emergence of the Edomite Kingdom in the Southern Levant.” Briefly, the authors of the present manuscript argue that there is insufficient evidence for an abrupt technological shift precipitated by the arrival of Egyptians.

I am generally in agreement that there are significant issues with the Ben-Yosef et al argument, certainly from the perspective of metallurgical technology, and also perhaps from the perspective of other evidence. It is productive to these discussions to see some pushback from among those working on archaeometallurgy in the Southern Levant. Given that the original paper was published in PLoSONE, it makes sense that the rebuttal should be published in the same venue to aid discoverability.

However, I think that the critique could be better formulated. There are methodological and framing issues with some of the critiques, others seem less relevant, while other avenues of critique are unexplored. Below, I note several areas for improvement.

This introduction is accepted.

150-160.

This framing of the critique, claiming that only “technological” evidence should be used in creating models of production system, not “archaeological and historical considerations,” is ineffective. The point seems to be that the focus should only be on analysis of direct production residues—slags, tuyères, and the like.

Most scholars of technology would agree that technological systems consist of much more than just the immediate techniques and behavioral sequences involving the act of production. In discussing organizational and management aspects, the authors themselves seem to acknowledge this. Thus, it seems problematic to claim that archaeological and historical considerations shouldn’t be included, with the goal of producing a “purely technological viewpoint.” Broader social, political, and economic considerations, revealed through analysis of the broader archaeological and historical evidence, are very much an important part of analyzing a technological system.

I suspect the authors realize all this, but the framing here could be improved.

It seems that the wording was not clear enough. It is not that building activities should be ignored, but rather - that changes in copper smelting technology should be evaluated based only on technological issues while building activities are not related to this aspect.

Thus, in spite of the fact that I fully accept the opinion of the first reader saying:

"I agree with your criticism here [against the paradigm of Ben-Yosef] – arguments about technological innovation should be based solely upon the evidence from studies of the technology",

I accept the concerns raised by the third reader and thus changed the text in order to mention the contra opinion too:

Lines 141 - 145:

The distinction between the Production Systems described above involved mainly archaeological considerations, some unrelated to the technology, such as building activity at Khirbet e-Nahas, as well as historical events [11]. However, in this unique case the focused should be given to technology development, mostly carried out by a limited number of personnel. Therefore, in our view, the developmental stages should be based only on technological considerations

Cluster Analysis Lines 161-165:

It seems the multivariate statistical techniques (161ff) are being done on averages, rather than on the primary data, which is available as supplementary information alongside the Ben-Yosef et al paper. This is potentially problematic. Aside from a justification of the use of averages, I’d also want to see some methodological discussion about applying the cluster analysis to a dataset with a combination of continuous chemical data (Cu%) and radiocarbon data, which are calibrated date ranges with probability density distributions that are not normal.

The Ben-Yosef et al. 2019 figure has its own problems (the use of only 1σ error bars for the date ranges, when 2σ is pretty much standard for reporting in archaeology).

The reviewer is correct.

Nevertheless, I think that Ben-Yosef et, al., should be praised for their important role in collecting all the available data in the field. However, they were not able to find a C14 result from an adjacent or associated location for each slag sample, and naturally, they had fewer C14 results than slag results, see Figs. S1, S2 and S3 [11].

So they had to use averages which seem to be the only practical solution possible in this case.

As a direct result, (lines: 151-156):

The residual copper concentrations in the slag samples from the Arabah sites and their average dates (as presented above in Figure 2, as shown in ref. [11]) were subjected to Cluster Analysis. The statistical procedure selected here was K-means [17], and the number of clusters were determined using the Elbow Method. The outcome is shown in Figure 3. It presents three clusters: Clusters 1 and 2 include samples that are related to the first Technological Phase (TP I), whereas Cluster 3 correlates to the second Technological Phase (TP II).

Continuation of the remark of the third reviewer:

Given these potential issues, I’m not sure that the statistical analysis is helpful or necessary to build the critique. What can and should be stated (without any need for statistical analysis), is that there is really no abrupt shift in the copper content of the slags as shown in the Figure 2 of this paper (Figure 3 of Ben-Yosef et al.). Lines 208ff makes this point quite effectively without recourse to cluster analysis. What you see is really a gradual linear trend, one that would likely appear even more gradual if the individual analyses were plotted rather than just averages. Visualizing this by recreating that figure with raw data rather than averages would actually quite useful, and would probably help the authors state their case. The data are available as supplementary information on the Ben-Yosef et al 2019 paper, so it should not be difficult to do. Furthermore, a critique of the Ben-Yosef et al. paper would do well to point out some potential limitations on the use of Cu content as a proxy for smelting efficiency.

It is true that the statistical analysis is not necessary for critiquing or denying Ben-Yosef et. al.’s, paradigm: a presentation of time vs. residual copper content will make it simpler.

Nevertheless, there are 4 important reasons for introducing the Cluster Analysis: presenting hidden information that is not included in the conventional presentation:

1. Cluster 1 reveals for the first time an unknown stage, or an "adaptation period", in which production efficiency had declined before it was increased. The reasons were explained in the text.

2. Cluster 3 differs from Cluster 2. This means that Cluster 3 represents a new and different stage of development (TP II).

3. The interface between Cluster 2 to 3 gives an assessment for estimating the time at which TP II was introduced.

4. Ben-Yosef et al, claim that PS I and PS II are two different stages in copper development. This is refuted by Cluster 2 - based only on technical issues - which demonstrates that, technically, PS I and PS II are part of the same development.

First, slags from pre-modern smelting slags are often quite heterogeneous. Because copper is heavier than the silicate matrix, it will tend to sink, so samples of slag taken from the top of a furnace slag cake will have less copper than a sample taken at the bottom. Many of these differences will be large compared to the average differences measured chronologically (a total range of about 1.5 wt% for the averages shown in figure 2.) For similar reasons, tap slags may differ in consistent ways from furnace slags. It may be worth looking into the Ben-Yosef et al 2019 supplementary data and methods information to see whether they took this into consideration. At the very least, it would be nice to have reassurance that they compared like to like (e.g. tap slags to tap slags).

--Second, slag composition is dependent significantly on the composition of the ore. Ben-Yosef et al mention this briefly, but it is worth discussing more fully. If the copper content of the ore used declined over time, which one might expect as individual mines are worked out, the copper content of the slags would also decease. Could the decreasing copper content of the slags be explained by the progressive working out of the mines exploited during the Iron Age? This is probably worth considering.

Discussion of these issues would strengthen the critique.

I have checked these intriguing issues in Ben-Yosef el al., 2019 and could not find any clue. However, in Ben - Yosef 2012 described 2 slag types:

(See: Ben-Yosef E, Shaar R, Tauxe L, Ron H. A New Chronological Framework for Iron Age Copper Production at Timna (Israel), Bulletin for the American Schools of Oriental Research 2012; 367: 31–71.)

"Type A consists of relatively large fragments (or slab) of Mn rich slag, sometimes more than 20 cm in diameter; the other (“Type B”) consists of small fragments of Fe-rich slag (cf. fig. 8); both are the result of tapping technology, and each corresponds to a distinct stratigraphic context. Type A, the Mn-rich slag, is present only in Layer I (in Area S only in Section S-2, Horizon 0) and represents the latest phase of copper production at the site, while the small Fe-rich fragments of Type B.”

260-273.

This discussion is not well supported. We have very little idea whether ancient smelters would have approached invention and experimentation in the same controlled way that R&D firms do today, only modifying one element of a procedure at a time. It is entirely possible that improvements were the result of post-hoc modification upon noticing an improved yield after an accidental modification of established procedure (i.e. copy errors, to use the term from evolutionary theory), rather than a goal-oriented, controlled experimental testing.

Please see my answer to the first reader on the same issue, lines 241 – 267.

In addition:

Statistically, it is unlikely to expect an "accidental modification" because of the numerous variables involved (ore type, ore quantity, flux type, flux quantity, temperature variation, air flow rate, oven type, tuyère location and inclination, etc...) that have to participate properly and optimally in such events.

If the event was spontaneous – it would be impossible to repeat it afterwards. On the other hand, thousands of random trials would be needed to cover only few successive "trial and error" attempts.

Therefore, as explained in the text, a "trial and error" with a single change in each testing stage, seems the preferred solution for a gradual development.

278-280.

This speculation about the copper-iron chunks goes too a bit too far. These copper iron chunks are referenced periodically, but they have not been studied well enough, especially with respect to their microstructure, to be able to say what process they come from.

I agree. The following was deleted.

However, it is possible that by-products denoted “chunks” (copper-iron mixtures with up to 70% iron), reported by Ben-Yosef [7, 278 pp. 711, 832] as "failed smelting cycles" are, in fact, slag remains from such unsuccessful trial and error episodes.

236-252

In the section “Advanced Techniques and Managerial capabilities in the Arabah industry” the authors do not do a good job of explaining how these processes (trial and error, scaling up, administration) differ from those proposed in the 2019 paper. After all, the 2019 paper does argue for a gradual improvement in Cu smelting technologies prior to the alleged sharp break corresponding with Sheshonq’s arrival. Trial and error, and scaling up can all be incorporated into the evolutionary model proposed by Ben-Yosef et al. It’s the “punctuated” part of the “punctuated equilibrium” concept where the models differ.

Ben-Yosef is not clear in this issue:

First he wrote that the locals were not involved in "any stages of "trial and error" or slow technological developments that might indicate local innovations" (Ben-Yosef 2010: 973). In his 2019 paper he discussed a gradual improvement, without specifying how it was achieved, but says: "After generations of internal efforts to better the technology—with limited success—the techno-social system was receptive of extraneous influences that facilitated the same cause" [p. 11].

My attempt was intended to explain and to reconstruct the developmental achievements. Also to reject the model of “punctuated equilibrium”

In order to improve the explanations in the technological and the managerial issues – the text was revised.

Please see: Advanced Techniques and managerial capabilities in the Arabah Industry, lines 227 - 266.

....In particular, the sub-section discussing the importance of managerial quality doesn’t help us distinguish whether the relevant administrative team was of local extraction or foreign.

A clarification was added in lines 384 386:

It would be very difficult to accept the possibility of an Egyptian involvement in the managerial level during the period between Early to Late Iron Age because such presence would have required the physical presence of qualified Egyptian personnel in the Arabah. However, such evidence has not been found.

380-381

This final sentence is important. Given the adjustments to the Timna chronologies over the last 15 years, how accurate are the chronological designations for the Sinai copper exploitations, particularly LBA/EIA?

Some assume: +25 years to -25 years

389-390.

It doesn’t quite follow how the slower adoption of iron in Egypt contributed to the copper deficiency. Further elaboration needed.

Line 412:

In addition, the process of iron dissemination to Egypt was slower than in the Levant, reaching its peak only in the second half of the first millennium BCE [29, pp. 167–168).

Therefore, the need for copper consumption inside Egypt during the Iron Age did not decrease, as it had in other areas throughout the Levant where copper was replaced by iron. All these factors most probably led to a copper deficiency in Egypt during the first millennium BCE

404-405

What is the evidence that copper technologies in the Arabah were superior to those in Egypt? Missing here is a discussion of local Egyptian copper smelting technologies for comparison.

This issue was revised and discussed above:

Our attempts to compare slag results from the Arabah with the slag results from Sinai and the Northern Eastern Desert - was found to be very difficult as only few results were reported.

The results found (n=16) were shown by Abdel-Motelib et al. (2012: 37, 40). The average residual copper was 18.8% (!) and the variation between the results was enormous.

As we didn't find more supporting evidence we decided not to publish these results.

Therefore, it is justified to change our wording on this subject.

I agree with Fantalkin and Finkelstein [39] who argued that the objective of Sheshonq I’s campaign was to "preserve and promote" the copper production in the Arabah, as, at that time, the Arabah source "must have been the major - if not only- source of copper for Egypt". As Sheshonq I was aware of the huge reduction in copper production in Egypt during the Iron Age [33; 34; 27] he decided to secure a steady copper supply from the Arabah to Egypt.

437-438.

Similarly: could it be that TPII technologies haven’t been found in Egypt or Sinai because the Arabah has been the subject of far more intensive and extensive archaeological research? Absence of evidence isn’t evidence of absence, especially given the disparities in archaeological research.

It is always possible that the TP II technology might be found in Sinai and/or in Egypt (or Saudi for that matter!). On the other hand, as was shown in the original text (lines 309-313) that the Egyptians succeeded to develop their own tuyère, most probably developed due to the same problems of clogging that existed in the Arabah:

See line 324:

The existence of reused tuyères indicates that the craftsmen at Bir-Nasib, Sinai faced the same clogging problem that existed in the small tuyère in the Arabah. However, they arrived at different solutions: (i) replacing the clogged nozzle with a new one, rendering it suitable for multiple operations and (ii) increasing the dimensions of the small tuyère, although not to the same large dimensions selected for the "long tuyère" in the Arabah (Table 2.).

Thus, as the Egyptians had their own solution, the possibility of finding TP II in Sinai or in Egypt are very low.

440-441

“No evidence for such a presence can be gleaned from the finds.” I would rephrase this, because proponents of an Egyptian presence could counter that the presence of a few overseers or a small military detachment might not leave a massive archaeological signature, and there are a handful of Egyptian objects as noted above. What the authors could reasonable say is that evidence for a substantial Egyptian presence is limited or equivocal.

As was shown, evidence for Egyptian presence is lacking. Egypt was “present” in the entire region during this time, is not taken as evidence for an actual presence. Nevertheless, the word 'substantial' has been added.

However, as noted above, no substantial evidence for such a presence can be gleaned from the finds.

Figure 5. I think “Slag buildup” is what is intended, not “slag built up.” And it’s Furnace not Furnance.

Thank you, these mistakes have been corrected.

Overall, I think that once these revisions have been made, this critique will form a useful contribution to the literature on technological change.

6. PLOS authors have the option to publish the peer review history of their article (what does this mean?). If published, this will include your full peer review and any attached files.

Do you want your identity to be public for this peer review? For information about this choice, including consent withdrawal, please see our Privacy Policy.

Reviewer #1: Yes: David Killick

Reviewer #2: No

Reviewer #3: No

---

## [Decision Letter · Decision Letter 1]

12 Nov 2021

Copper Technology in the Arabah during the Iron Age and the Role of the Indigenous population in  the Industry

PONE-D-20-15801R1

Dear Dr. Luria,

We’re pleased to inform you that your manuscript has been judged scientifically suitable for publication and will be formally accepted for publication once it meets all outstanding technical requirements.

An invoice for payment will follow shortly after the formal acceptance. To ensure an efficient process, please log into Editorial Manager at http://www.editorialmanager.com/pone/ , click the 'Update My Information' link at the top of the page, and double check that your user information is up-to-date. If you have any billing related questions, please contact our Author Billing department directly at authorbilling@plos.org .

If your institution or institutions have a press office, please notify them about your upcoming paper to help maximize its impact. If they’ll be preparing press materials, please inform our press team as soon as possible -- no later than 48 hours after receiving the formal acceptance. Your manuscript will remain under strict press embargo until 2 pm Eastern Time on the date of publication. For more information, please contact onepress@plos.org .

Kind regards,

Anwar Khitab

Academic Editor

PLOS ONE

Additional Editor Comments (optional):

Reviewers' comments:

Reviewer's Responses to Questions

**Comments to the Author**

1. If the authors have adequately addressed your comments raised in a previous round of review and you feel that this manuscript is now acceptable for publication, you may indicate that here to bypass the “Comments to the Author” section, enter your conflict of interest statement in the “Confidential to Editor” section, and submit your "Accept" recommendation.

Reviewer #2: All comments have been addressed

2. Is the manuscript technically sound, and do the data support the conclusions?

Reviewer #2: Yes

3. Has the statistical analysis been performed appropriately and rigorously?

Reviewer #2: Yes

4. Have the authors made all data underlying the findings in their manuscript fully available?

Reviewer #2: Yes

5. Is the manuscript presented in an intelligible fashion and written in standard English?

Reviewer #2: Yes

6. Review Comments to the Author

Reviewer #2: (No Response)

7. PLOS authors have the option to publish the peer review history of their article (what does this mean? ). If published, this will include your full peer review and any attached files.

**Do you want your identity to be public for this peer review?** For information about this choice, including consent withdrawal, please see our Privacy Policy .

Reviewer #2: No

---

## [Editor Report · Acceptance letter]

17 Nov 2021

PONE-D-20-15801R1

Copper Technology in the Arabah during the Iron Age and the Role of the Indigenous Population in the Industry

Dear Dr. Luria:

I'm pleased to inform you that your manuscript has been deemed suitable for publication in PLOS ONE. Congratulations! Your manuscript is now with our production department.

If your institution or institutions have a press office, please let them know about your upcoming paper now to help maximize its impact. If they'll be preparing press materials, please inform our press team within the next 48 hours. Your manuscript will remain under strict press embargo until 2 pm Eastern Time on the date of publication. For more information please contact onepress@plos.org .

If we can help with anything else, please email us at plosone@plos.org .

Kind regards,

on behalf of

Dr. Anwar Khitab

Academic Editor

PLOS ONE